



# Spatial Distribution of the Persistent Organic Pollutants across the Tibetan Plateau and Its Linkage with the Climate Systems: Five Year Air Monitoring Study

5    Xiaoping Wang[1,2*],Jiao Ren[1],Ping Gong[1,2],Chuanfei Wang[1],Yonggang Xue[1],Tandong Yao[1,2], Rainer Lohmann[3]

[1]Key Laboratory of Tibetan Environment Changes and Land Surface Processes, Institute of Tibetan Plateau Research, Chinese Academy of Sciences (CAS), Beijing, 100101, China

[2]CAS Center for Excellence in Tibetan Plateau Earth Sciences, Beijing, 100101, China

[3]Graduate School of Oceanography, University of Rhode Island, Narragansett, Rhode Island 02882-1197, USA

* **Corresponding author address:** E-mail: wangxp@itpcas.ac.cn

**Abstract.** The Tibetan Plateau (TP) has been contaminated by persistent organic pollutants (POPs), including legacy organochlorine pesticides (OCPs) and polychlorinated biphenyls (PCBs) through atmospheric transport. The exact source regions, transport pathways and time trends of POPs to the TP are not well understood. Here XAD-based passive air samplers (PAS) were deployed at 16 Tibetan background sites from 2007 to 2012 to gain further insight into spatial patterns and temporal trends of OCPs and PCBs. The southeastern TP was characterized by dichlorodiphenyltrichloroethane (DDT) -related chemicals delivered by Indian Monsoon air masses. The northern and northwestern TP displayed the greatest absolute concentration and relative abundance of hexachlorobenzene (HCB) in the atmosphere, caused by the westerly-driven European air masses. The interactions between the DDT polluted Indian monsoon air and the clean westerly winds formed a transition zone in central Tibet where both DDT and HCB were the dominant





chemicals. Based on 5-year of continuous sampling, our data indicated declining concentrations of HCB and hexachlorocyclohexanes (HCHs) across the Tibetan region. Inter-annual trends of DDT class chemicals, however, showed less variation during this 5-year sampling period, which may be due to the on-going usage of DDT

in India. This paper demonstrates the possibility of using POPs fingerprints to investigate the climate interactions and the validity of using PAS to derive inter-annual atmospheric POPs time trends.

## 1 Introduction

Organochlorine pesticides (OCPs) and polychlorinated biphenyls (PCBs) are classified and regulated as persistent organic pollutants (POPs) under the Stockholm Convention (Patterson et al., 2009). Although the extensive usage of these pollutants has been forbidden for several decades, they are still ubiquitous in the global environment and the atmosphere plays a vital role in their global dispersal (Corsolini et al., 2002;Harner

et al., 2006;Jones and De Voogt, 1999).

The spatial distribution of POPs is subject to three factors. On the one hand, the global POPs spatial patterns strongly related to the POPs emissions with higher levels appeared in urban and agricultural sites where POPs had undergone extensively historical or current usage (Harner et al., 2004). On the other hand, long range

atmospheric transport (LRAT) is responsible for the global transport of POPs, resulting in relatively high burden of POPs in remote area (Aulagnier and Poissant, 2005;von Waldow et al., 2010). Thirdly, the global pattern of POPs could be reassigned by the secondary emissions of POPs which is a result of "multi-hopping" between air and land/sea surfaces (Lohmann et al., 2007;Nizzetto et al., 2010). The last two factors are

strongly influenced by climate variations (temperature, wind, precipitation, and others).

Temperature difference is the important factor that determines the difference in POPs concentrations in air (Lamon et al., 2009;Pelley, 2004). Higher temperatures can drive





increased volatilization emissions of POPs, and enhance the POPs transport (Dalla Valle et al., 2007;Lamon et al., 2009). The wind flows associated with the climate fluctuations can also exert influence on the transport strength and pathway of POPs (Pelley, 2004). Higher wind speeds can result in more efficient intercontinental transport. The Arctic Oscillation (AO), the North Atlantic Oscillation (NAO) and the Pacific North American (PNA) pattern are three major sources of climate variability in north Hemisphere. During the positive phase of NAO, stronger westerly winds blow across the North Atlantic Ocean, enhancing the transport of POPs from the Canadian Prairies to Great Lakes region (Ma et al., 2004a;Ma et al., 2004b). When the PNA pattern intensified, the southwesterly flow along the west coast of Canada increased and gave rise to a poleward transport of POPs to the Arctic (Gao et al., 2010;Ma et al., 2004b).

Wet deposition is the important way to scavenge pollutants from air. Modeling results suggested that maximum POPs removal occurred in places with extensive wet deposition (i.e., the Intertropical Convergence Zone (ITCZ) and the region with plenty of snow) (Jurado et al., 2005). With climate change, ITCZ shifts with seasons and extreme snowy weather events frequently occur (Yancheva et al., 2007;Bednorz, 2002), which will also influence the global distribution of POPs.

Similar to the north and south poles, the Tibetan Plateau (TP) is a region of particular interest to scientists studying both climate change and POPs contamination. The average elevation of the TP is 4700m, which acts like a "wall", and splits the mid-altitude westerly into two currents (Wang et al., 2010). Moreover, the TP has an area of $2.5 \times 10^6$ km$^2$, which enhances the land-sea thermal contrast and strengthenes the Indian monsoon (Wang et al., 2010). Climate of the TP displayed spatial difference with the north/west plateau controlled by westerlies and the south/east is dominated by the Indian monsoon (Schiemann et al., 2009). Contaminant transport over the TP is therefore complicated due to the different climate systems (transport pathways) and different source regions. For example, different perfluoroalkyl acids composition



profiles were observed for snow-cores from different regions of the TP (Wang et al., 2014). The seasonal variation of DDTs in southeastern TP is synchronous with the seasonality of the Indian Monsoon, suggesting that the monsoon-driven transport of POPs to the TP is continuous and sustained (Sheng et al., 2013). Although connections between climate circulation and pollutants transport have been observed, the above studies are limited to few sampling points. Actually, the interactions between the Westerlies and the Indian Monsoon are of great concern given they will influence the moisture, heat and anthropogenic forcing on regional and global scale. Thus, it is critical to conduct regional air sampling across the TP and investigate the linkage between spatial patterns of POPs over the TP and the interactions between climate systems.

Taking advantage of the Tibetan Observation and Research Platform (TORP) (Ma et al., 2008), a large-scale and long-term (2007-2012) atmospheric POPs monitoring program across the TP was conducted and the XAD-2 based passive air samplers (PAS) were deployed. The result of the first year (2007-2008) has been given previously (Wang et al., 2010). In this study, all 5-year data (2007-2012) were gathered to get more reliable spatial POPs patterns and to investigate the role of the climate systems in forming such patterns. The temporal trends of atmospheric POPs concentrations during this 5-year period were also investigated. A better understanding of these processes will contribute to determining how the global climate systems will affect the spatial distribution of contaminants and how background POPs levels over the TP vary with time increase.

## 2 Materials and Methods

**2.1 Sampling Sites and Programs.** An important objective of this study was to improve the knowledge of spatial patterns of POPs in background air across the TP using PAS. Therefore, 16 background sites were selected in the context of LRAT and represented a good spatial coverage of the TP [see the Supplementary Information (SI), Figure S1]. Sampling sites located from Muztaga (38°N) in the north to Mt. Everest (28°N) in the south, and from Muztaga (75°E) in the west to Chayu (97°E) in the east;





altitudes of sampling sites ranged from 500 to 5200 m (Table S1). Due to its high capacity and continuous accumulating over year-round exposure for POPs, XAD-PAS were chosen in this study.[24] Consecutive five-year (July 2007 to August 2012) air monitoring program were conducted and XAD-PAS were deployed for five one-year

periods to yield annually averaged concentrations of POPs. Duplicate samples were deployed at 6 sites (Lhasa, Xigaze, Lhaze, Mt. Everest, Lulang and Namco), and field blanks were taken at Golmud, Qamdo, Lulang, Lhasa, Mt.Everest and Gar for every sampling year. The Table S1 also gives details of the sampling program including the date of sampling collection and the brief introduction about the sites. An active air

sampler (AAS) was additionally co-deployed with XAD-PAS at Lulang (Figure S1) to evaluate possible differences between PAS and AAS. The sampling period of AAS was from November 2008 to September 2011, and details about the sampling AAS program were reported in previous study (Sheng et al., 2013) and also provided in Text S1.

**2.2 Sample Preparation, Extraction and Analysis.** Prior to deployment, XAD resin

was Soxhlet extracted using in turn methanol, acetonitrile, and dichloromethane (DCM). The XAD resin (60 mL of wet XAD in methanol) was transferred to a precleaned stainless steel mesh cylinder and dried in a clean desiccator. Dry cylinders were sealed in an airtight stainless steel tube with Teflon lids. Upon completion of the sampling, all samples (XAD-cylinders) were stored at -20°C until extraction. Chemical extraction,

cleanup and details of gas chromatographic temperature are given in Text S2. The following compounds were measured and quantified: PCB 28, 52, 101, 138, 153 and 180; HCB; o,p'-DDE, p,p'-DDE, o,p'-DDT and p,p'-DDT; α-HCH, β-HCH, γ-HCH and δ-HCH.

**2.3 Quality Assurance/Quality Control (QA/QC).** All analytical procedures were

monitored by strict QA/QC measures. The blank concentrations and the corresponding methods detection limits (MDLs) are given in Table S2. MDLs were derived as the mean field blank concentration plus 3 times of its standard deviation. In the present study, MDLs values ranged between 0.04 and 1.2 ng per sample for OCPs,



and between 0.10 and 0.32 ng per sample for PCBs (Table S2). The recoveries were between 70% and 106% for 2,4,5,6-tetrachloro-m-xylene (TCmX) and between 72% and 114% for decachlorobiphenyl (PCB-209). The full dataset of the POPs concentrations (in unit of ng per sample) over the five sampling years is available as

Table S3. All reported values were blank-corrected but not corrected for the recoveries. If the concentration of a compound after blank correction was below the MDL, the concentration was substituted with 1/2 MDL (Table S2). Duplicate PAS were deployed to check the repeatability and the results showed the average relative deviation of concentrations between duplicates is generally low, which ranged from 9%

to 18% for different compounds (Table S4). Uncertainties for chemical analysis were in the range of 25–35% and reasons for the uncertainties were discussed in Text S3.

**2.4 Sampling rate of XAD-PAS and its uncertainties.** Sampling rate (R, $m^3$/day/PAS) was estimated using the empirical equation provided in the previous study (Wang et al., 2010), $R = 0.16 \times \frac{T^{1.75}}{P} - 2.14$ (eq 1). The advantage of this

empirical equation is that one can easily deduce the R of XAD-PAS by using the general parameters of temperature (T, K) and pressure (p, hpa) in a sampling site (R of each individual site was provided in Table S5). After obtaining the sequestered chemical amount per sampler and R, the PAS results can be derived to yield volumetric air concentrations ($pg/m^3$, Table S6), which is essential to conduct a direct

comparison with AAS results. In this study, both XAD-PAS and AAS were deployed in Lulang and details about the data comparison were provided in Text S4. Good agreement was found between the derived PAS concentrations and AAS results. This indicated that the sampling rate deduced by equation (1) is relatively reliable. However, uncertainties associated with R can be from variable environmental

conditions, such as turbulent wind and the air temperature inside the sampler shelter which can be different from the temperature recorded by weather stations.

**2.5 Statistical method.** Cluster analysis is a multivariate procedure that establishes the relationship among objectives (here samples). The similarity among samples is



usually defined as the Euclidean distance between the samples. Cluster analysis divides a set of samples into different groups (clusters), and each sample unambiguously falls into one of the groups. The samples in the same group are more similar to each other due to the short Euclidean distance than to those in other clusters. The dendrogram of the cluster analysis finally provides visual evaluation on the relationships between different samples, i.e. close or distant. Using the XLSTAT software (Addinso, France), we applied cluster analysis (agglomerative hierarchical clustering) to 5-year Tibetan PAS datasets that have been obtained above. Data from XAD-resin based PAS networks are usually reported in units of ng per sampler. However, cluster analyses were performed on compositional data, not on absolute concentrations. More details about the methodological issues are available in the previous literature (Liu and Wania, 2014).

### 3 Results and discussion

**3.1 Concentrations of POPs in the Atmosphere of TP.** The full dataset for the POPs concentrations (ng/sample) in individual sampling year are available as Table S3 and the average values were provided in Table S7. The dominant chemicals in the atmosphere of TP were HCB followed by o,p'-DDT and p-p'-DDT, with the average concentration up to 30.0, 13.4 and 11.1 ng/sample, respectively (Table S7). Nine chemicals (α-HCH, γ-HCH, HCB, o,p'-DDE, p,p'-DDE, o,p'-DDT, p,p'-DDT, PCB 28 and PCB 52) were regularly detected in samples, while highly chlorinated PCBs such as PCB 180 were never detected in samples (Table S3). To further identify the POPs contamination level on the global scale, a comparison was performed between the current study and the previous Global Atmospheric Passive Sampling (GAPS) study which also relied on the XAD-PAS (Shunthirasingham et al., 2010). The levels of HCB in the Tibetan atmosphere ranged from 2.5 to 60.7 ng/sample (Table S3), which is higher than the background areas of Asia (4.6-27.4 ng/sample) and South America (8.9-17.8 ng/sample), but similar to those in Europe (12.6-78.1 ng/sample) (Shunthirasingham et al., 2010). The level of α-HCH (0.1-16.4 ng/sample) and γ-HCH (0.1-18.7 ng/sample) is comparable to the remote stations in North America (Below



detection limit-15.3 ng/sample) and Africa (1.4-18.7 ng/sample), respectively (Shunthirasingham et al., 2010). Regarding the PCBs, its value ranged from 0.1 to 2.8 ng/sample in this study, which is much lower than those of Arctic Region (1.7-14 ng/sample) (Shunthirasingham et al., 2010). These comparisons indicate that POPs

concentrations in the atmosphere of TP are akin to those observed in the background sites of the world, indicating the remoteness of TP. However, it is worth noting that the DDTs levels of TP (0.1-38.9 ng/sample for o.p'-DDT and 0.9-22.7 ng/sample for p-p'-DDT, Table S3) were considerably higher than other remote regions (mostly not detected) (Shunthirasingham et al., 2010), which may suggest closer proximity of the

TP to the DDT source regions.

The derived volumetric concentrations (Table S6) were further averaged (Table S8) and compared with the values reported for surrounding countries of TP (Table 1). By comparison, the levels of OCPs and PCBs in TP air were all much lower than the neighboring regions, such as India (Zhang et al., 2008), Nepal (Gong et al., 2014) and

Pakistan (Syed et al., 2013). Even for the background sites of India, it is one order of magnitude higher than those of TP (Table 1) (Zhang et al., 2008). POPs produced and consumed in these highly contaminated countries are potential to undergo LRAT and contaminate the pristine environment of TP, especially under the favorable climatic conditions. Furthermore, the concentration range of POPs across the TP was large

(Table 1), which imply some distinctions among different parts of TP and it is necessary to figure out the factor(s) causing this spatial variability.

**3.2 Spatial Distribution of POPs across the TP.** The spatial distribution of DDTs, HCHs, HCB and PCBs has been reported in our previous study, based on one-year (2007-2008) of data (Wang et al., 2010). This first-year study indicated some "clues"

on the possible transport and distribution features of POPs across the TP. Yet a single year is insufficient in comprehensively understanding the POPs' spatial patterns. The integration of the long-term monitoring data is therefore needed. For each sampling site, the mean values of air concentrations over multi-year monitoring are more



representative than concentrations for a single year. Here, the mean values of each sampling site were graphically presented in Figure 1 to address the spatial distribution patterns of various POPs on the Tibetan Plateau.

The spatial distribution of DDTs shows a decreasing gradient from the southeast to the northwest of TP (Figure 1). Higher values of DDTs were found at Chayu (39.0 pg/m³), Bomi (35.2 pg/m³) and GBJD (31.3 pg/m³), which are located around the Yarlung Tsangpo Grand Canyon – a channel for receiving the pollutants from the south Asia. Similarly, higher levels of HCHs were also observed at Bomi (10.8 pg/m³), and GBJD (6.3 pg/m³). Previous studies suggested that the Yarlung Tsangpo River valley is considered a "leaking wall" that contaminates the southeast Tibetan Plateau (Sheng et al., 2013;Wang et al., 2015). Given the Indian Monsoon is the distinct climate of the southeast Tibetan Plateau, and DDTs and HCHs are the dominants POPs in Indian atmosphere (Zhang et al., 2008;Gioia et al., 2012), the monsoon-driven LRAT of POPs is thus the possible reason why higher DDT/HCH contaminations occurred in southeast of Tibetan Plateau (Sheng et al., 2013). Besides the LRAT, scattered usage of HCHs cannot be ruled out. For example, high levels of HCHs were also found in Naqu during this 5-year periods. Naqu is located closer to the central plateau and is an agriculture and pasturage interlaced zone. Thus, the scattered HCH usage in Naqu might be possible. The spatial pattern of HCB is opposite to those observed for DDTs and HCHs, with relatively high concentrations occurred in north and west TP (Figure 1). In addition, all sampling sites in this study displayed low concentrations and a uniform spatial distribution pattern for PCBs (Table S8 and Figure 1). This is a typical feature of a remote region and indicated that there is very limited primary emission of PCBs in the TP.

**3.3 Dose the soil-air exchange (secondary source) affect the spatial pattern?**

Due to the remoteness of the Tibetan Plateau, both atmospheric transport and regional re-evaporation can be two important vectors that influence the spatial distribution pattern of POPs. Soil has been reported as a major environmental reservoir of POPs.





Cabrerizo et al. (2011a) found that air-soil exchange controlled atmospheric concentration of OCPs in background region. Sampling air and soil at the same sites and the determination of soil and ambient air fugacities can provide quantitative evidence to test if soil is volatizing POPs to the atmosphere (the direction of soil-air exchange); how much of POPs will be evaporated (the fluxes of exchange); and if soil evaporation contribute to the observed spatial pattern.

In this study, soil data used for fugacity calculations were taken from a published study (Wang et al., 2012) in which the soils were collected from 2007. Due to its heterogeneity and slow rate of change, chemical concentrations in soil seldom change with time (Schuster et al., 2011). Therefore, we assume the soil concentration of OCPs and PCBs during the 5-year (2007-2012) sampling period were constant. Relevant equations regarding calculation of the air ($f_a$) and soil fugacity ($f_s$) were provided in supporting information (Text S5). The soil-air fugacity fraction (*ff*)

$$ff = f_s/(f_s + f_a) \quad \text{(eq 2)}$$

was then assessed for 13 sampling sites where both air and soil data sets were available (Table S9). A *ff* = 0.5 indicates that soil fugacity and air fugacity are same, and the compound is at equilibrium. Due to uncertainties, fugacity fractions between 0.3 and 0.7 were not considered to differ significantly from equilibrium (Harner et al., 2004;Li et al., 2010). Figure S2 showed the box-and-whisker plot of the *ff* for different chemicals. From Figure S2, HCB, PCB-28, PCB-52, and α-HCH showed mean *ff* values slightly higher than 0.7, while mean *ff* values of DDTs were similar to or lower than 0.3. This suggested the air-soil exchange of volatile compounds (PCBs, HCB and a-HCH) was on the verge of equilibrium but tended to volatilization; but the air-soil exchange of less volatile compounds was prone to deposition.

γ-HCH showed a small range and a high mean *ff* (close to 1) falling within the interquartile range, indicating a stable volatilization from soil to air (Figure S2). Meanwhile, higher volatilization fluxes of γ-HCH (up to 2.8 ng/m$^2$/h) were also found (equations used for calculating the volatilization fluxes were given in Text S5 and



fluxes values were provided in Table S10 and Figure S3). However, HCHs are generally regarded as "swimmers", which can easily be scavenged by precipitation (Lohmann et al., 2007) and thus its out-gassing from soil may not strongly influence its spatial distribution pattern. With regard to other chemicals (PCB-28, PCB-52, α-HCH

and HCB), their $ff$ displayed large overlap with equilibrium range (Figure S2) and only lower evaporation fluxes were found (Figure S3). This indicated the re-volatilization of these chemicals may not be a great contributor to their spatial distribution pattern. Given the DDT-class chemicals showed a deposition status (Figure S2), their spatial distribution patterns was therefore less influenced by secondary emissions.

In order to test if soil volatilization truly controlled the chemical's atmospheric occurrence, fugacities of chemicals in soil were correlated with those in air (Figure S4). We found that the individual chemicals escaping from soil (soil fugacity) were not statistically correlated with their ambient air fugacity (with very low $R^2$ and p > 0.1, Figure S4). This suggested that the out-gassing from Tibetan soil was not able to exert a

clear control on the atmospheric occurrence of OCPs and PCBs. Although HCB is an example chemical ("multi-hopper") which atmospheric concentrations is significantly influenced by the secondary-emission (Bailey, 2001), there was not a significant correlation obtained (Figure S4). This implied that the LRAT of chemicals instead of secondary sources were more important for the atmospheric concentrations and

distribution patterns of POPs over the TP.

**3.4 The spatial differences in POP sources and transport.** Although the spatial POPs concentrations were of primary importance, the relative composition of POPs on different spatial scales is also of interest. Cluster analysis was conducted to test if these different samples from different sampling years displayed similarity (samples with

similar POPs compositions are influenced by the similar source or similar transport pathways/climate systems), and can be grouped according to their POPs "fingerprints".

In the present study, we summed up levels of frequently detected compounds in each sample and normalized the level of individual chemical by this sum to yield a relative



fraction (%, Table S11). The results of cluster analysis are presented in Figure 2. In this study, all samples were classified into 3 groups: Group 1, Group 2 and Group 3 from the left to the right (Figure 2).

The Group 1 comprised 28 samples (Table S12). If data of the observation sites through most of sampling years can be grouped in the same cluster, this suggests that these sites have the real and consistent similarity. However, for the sites in which only one or two years of data can be included, these sites are doubtful sites and will not be considered as the representative of the group. In group 1 of this study, all samples from Chayu (one sample from this site got lost and in total 4 samples were harvested), 80% of samples (4 in 5 sampling years) from Lulang, and 60% of samples (3 in 5 sampling years) from GBJD, Bomi and Rawu were clustered in this group (Figure 2, Table S12, 13). All these sites (Chayu, Lulang, GBJD, Rawu, and Bomi) located in southeast TP and could be regarded as the representative sites of this group. Although some scattered samples e.g. Gar 1, Xigaze 1, 3 and Lhasa 3, 5 were clustered in this group, they were not likely the typical sites due to their lower frequency in this group. The dominant compounds of Group 1 were DDT-related chemicals (Figure 2), which contributed 56.5% (34.9-79.1%) to the total POPs. Taken the results of spatial distribution and cluster analysis together, both higher concentrations and higher proportions of DDT class chemicals were found in southeast TP. Transport of the POPs strongly relied on the Indian monsoon which make major impact on the plateau's south side (Sheng et al., 2013) but is blocked from going further north. Based on the similarity of sampling sites where shared the same Indian source, the main contributions of cluster analysis helped to identify the spatial influential coverage of the Indian monsoon over the TP.

Regarding the Group 2, it included 17 samples (Table S12); all Muztagata samples (3 samples were harvested), 80% of Golmud samples and 60% of Gar samples were grouped into this cluster (Table S12, 13). In this group, the representative sites (Muztaga, Golmud and Gar) were from north and west TP. HCB occupied the overwhelming majority in this group, which accounted for 77.1% (69.0-88.4%) of the





total POPs (Table S12). Meanwhile, higher HCB absolute concentrations were also observed for these sites (Table S8). It should also be noted that climate of north and west TP is mainly influenced by westerlies and their upwind POPs source regions include Europe and central Asia (Wang et al., 2014;Xu et al., 2009). Generally, the larger the percentage of HCB at a site, the cleaner it is (Liu and Wania, 2014). The GAPS study found that HCB is dominantly and uniformly distributed in the European air (Shunthirasingham et al., 2010). Similar results have also been observed by EMEP monitoring program (Halse et al., 2011). Therefore, Group 2 of the present study could reflect a regional fingerprint of clean European air.

The remaining 30 samples were classified into Group 3 and samples from Lhaze, Naqu, Lhasa, Everest and Qamdo had higher frequency (>60%,Table S13) in this group. Due to the possible local contamination of Naqu (see discussion above), Naqu was not included as a representative site of this group. For Group 3, the dominant chemicals were HCB followed by DDTs, comprising 54.6% (29.3-67.6%) and 24.7% (5.6-43.3%) of the total POPs, respectively and the representative sampling sites were mainly located on the central TP. This indicated the effective regional atmospheric mixing likely happened on the central TP where both Indian and European sources could be seen. Samples of 3 sites (Xigaze, Namco and Saga) did not specifically fall in any group but were scattered distributed among these 3 groups. In part, this may be caused by the bias originating from the laboratory analysis uncertainties. On the other hand, as these sites are in closer proximity with other sites in group 3 and are all from the central TP with relative uniform geography, Xigaze, Namco and Saga were more likely belong to the group 3. Data of duplicate samples were brought into the cluster analysis to test if the uncertainties of laboratory analysis could affect the clusters results (Figure S5). Nineteen pairs of duplicates appeared in the same group; only 4 of 23 pairs of duplicates were dispersed in different groups. This suggested that cluster analysis extracted real difference/similarity between sampling sites, instead of analytical variability.



Based on the classification derived from the cluster analysis, the whole Tibetan Plateau could be divided into 3 parts with three distinct POPs fingerprints (Figure 3): one reflecting the Indian monsoon air mass (DDTs, southeast TP), one reflecting the clean westerly air mass (HCB, northwest TP) and one that is just the mixture of this two air mass (mixed DDT and HCB, central TP). The formation of this spatial pattern can be attributed to the direct influences and interactions between different atmospheric circulations (Indian monsoon and westerly winds). From Figure 3, we roughly assigned the region, south of $30°N$ and east of $92°E$, as the monsoon region; the region, north of $35°N$, as the westerly domain; and then the region located in between these two domains (from 30 to $35°N$, and west of $92°E$) was regarded as the transition domain, which is under the control of a shifting climate between Indian monsoon and westerly.

Basically, precipitation oxygen isotope ratio ($\delta^{18}O$) is an integrated tracer of the atmospheric processes, which has been employed to investigate the interaction between the westerlies and Indian monsoon on the TP (Tian et al., 2007). Based on long-term observations of precipitation $\delta^{18}O$ on 20 stations of the TP, Yao et al. (2013) found that there is a transition domain located in between the westerly region and monsoon region. This is akin to our POP fingerprints pattern. Thus, outcomes of this study recommended that the POPs fingerprint could also act as a tracer like $\delta^{18}O$ to estimate the interactions between climate systems. As compared with precipitation collection which is expensive and laborious (every single rain or snow event should be collected), PAS is cheap and simple. If XAD-PAS can be employed with sufficient spatial resolution and coverage, the POPs fingerprints difference across the TP will be more distinct and clear boundaries between monsoon region, westerly region and transact region can be captured. Especially for the remote region like the TP, POPs fingerprints obtained by the PAS would certainly help to understand the realistic synoptic atmospheric patterns. So far, researchers had paid attention to how climate change is affecting POPs cycling (Bustnes et al., 2010;Dalla Valle et al., 2007). However, the opposite way of thinking allows us to use POPs fingerprints as possible chemical tracers to track the climate dynamics and global pollution diffusion events.





**3.5 Temporal Trends.** Long-term air monitoring of POPs can also provide temporal patterns which can be used to evaluate the effectiveness of the regional regulations on POPs. On the one hand, the concentrations observed over this 5-year period can be used as a benchmark for the future comparisons. On the other hand, given the TP is the hinterland abutted by Central/East/South Asia, the temporal patterns of POPs can also be used to test whether there are evidences of decreasing concentrations for this wide Asiatic region. Therefore, the inter-annual variation of the atmospheric POPs was given in Figure 4 using the box-and-whisker plot. The decreasing concentrations of α-HCH, γ-HCH, PCBs and HCB were observed (Figure 4). Regression analysis revealed that the concentration declining of these compounds with sampling year (from 2008 to 2012) were significant (Figure S6), which is in good agreement with the result of the GAPS study (Shunthirasingham et al., 2010). This suggested the effectiveness of Stockholm Convention in reducing the emissions of these substances in Asia countries to background atmosphere. As compared with other compounds, the levels of two parent DDTs went down-and-up during the 5-year sampling period (Figure 4) and the plots of DDT concentration versus sampling year did not show significant correlations (Figure S6). Randomized block ANOVA was further performed to test differences in the concentrations of DDTs among the 5 years sampling (2007 to 2012) in all sampling sits. The p-p'-DDT and p-p'-DDE concentrations in southeast TP (monsoon region) did not differ significantly among the 5-year of sampling (Table S14). As mentioned above, the southeast TP is receptor region of India source, the observed temporal pattern of DDT suggested that the regulation of DDTs in India might be less effective (Sharma et al., 2014). This suggested that on-going DDT usage in the aspect of health and epidemic prevention and the illegal DDT application for agricultural purpose in India needed to be better controlled and regulated.

## 4. Conclusion and implication

This study confirms that for a remote region, the spatial distribution patterns of POPs are closely related to the variations and interactions of climate systems. This study

also highlights that POPs' fingerprints can be used as chemical tracer to track the interactions of climate systems. This is of great significance as it indicated a simple and cost-effective PAS can yield valuable data on the realistic synoptic atmospheric interactions. Take into account of the close connections between climate fluctuations

(AO, NAO, PNA and ENSO) and POPs levels/fingerprints, spatial and temporal POPs variations reflected by PAS technique can provide extra evidences for understanding the process of climate change. The results obtained from this study also highlighted the feasibility of PAS to serve in identification of inter-annual trends of POPs. Long term air monitoring of POPs using PAS samplers can therefore be used to evaluate the

effectiveness of the Stockholm Convention.

Results obtained from our investigation emphasized the need for performing more studies to better understand the secondary emission of POPs over the TP. Fugacity sampler (Cabrerizo et al., 2011b) was therefore recommended for the future studies due to it can provide accurate air-soil fugacity gradients. More studies should also be

conducted to reduce the uncertainty of PAS sampling rate and get the relatively accurate air concentration, which is essential for global comparison and discerning the time trends.

## 5. Acknowledgment

This study was supported by the National Natural Science Foundation of China

(41222010 and 41071321) and Youth Innovation Promotion Association (CAS2011067). Wang X. acknowledged the staff at the Southeast Tibet Observation and Research Station for the Alpine Environment for helping with field sample collection. Details about sampling collection, analysis, detection limits, concentrations, air−soil gas exchange calculations, and regression analysis of the time trend were

provided in Supporting information.

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





**Table 1. Concentration results of this study and the comparison with literature values reported for the surrounding countries (pg m$^{-3}$)**

| Sampling site | TP (Whole TP, this study) | India (urban) | India (rural) | India (background) | Pakistan (Punjab Province) | Nepal (southern slope of Himalaya) |
|---|---|---|---|---|---|---|
| year | 2007-2012 | 2006 | 2006 | 2006 | 2011 | 2012 |
| Sampler type | XAD-PAS | PUF-PAS | PUF-PAS | PUF-PAS | PUF-PAS | XAD-PAS |
| α-HCH | 3.8(0.1-17.7) | 451(22-1691) | 53(12-167) | 25(20-31) | 19(6.4-29) | |
| γ-HCH | 1.6(0.1-20.1) | 909(135-3562) | 174(31-437) | 61(34-100) | 20(5.4-45) | |
| HCB | 17.8(3.0-85.0) | | | | 33(13-76) | 234(128-416) |
| o,p'-DDE | 0.8(0.03-8.7) | | | | 63(12-240) | 10.6(BDL-41) |
| p,p'-DDE | 2.2(0.1-18.1) | 554(26-2061) | 81(15-282) | 13(6-19) | 79(4.2-290) | 154(17-597) |
| o,p'-DDT | 7.9(0.1-44.5) | 268(23-620) | 88(BDL--307) | 52(BDL-78) | 30(3.3-77) | 159(33-509) |
| p,p'-DDT | 4.4(0.1-26.1) | 110(2-249) | 79(3-387) | 25(9-45) | 34(6.0-66) | 125(21-456) |
| ∑ PCBs | 0.8(0.1-3.9) | | | | | 26.9(3.2-78.5) |
| Reference | | (Zhang et al., 2008) | (Zhang et al., 2008) | (Zhang et al., 2008) | (Syed et al., 2013) | (Gong et al., 2014) |





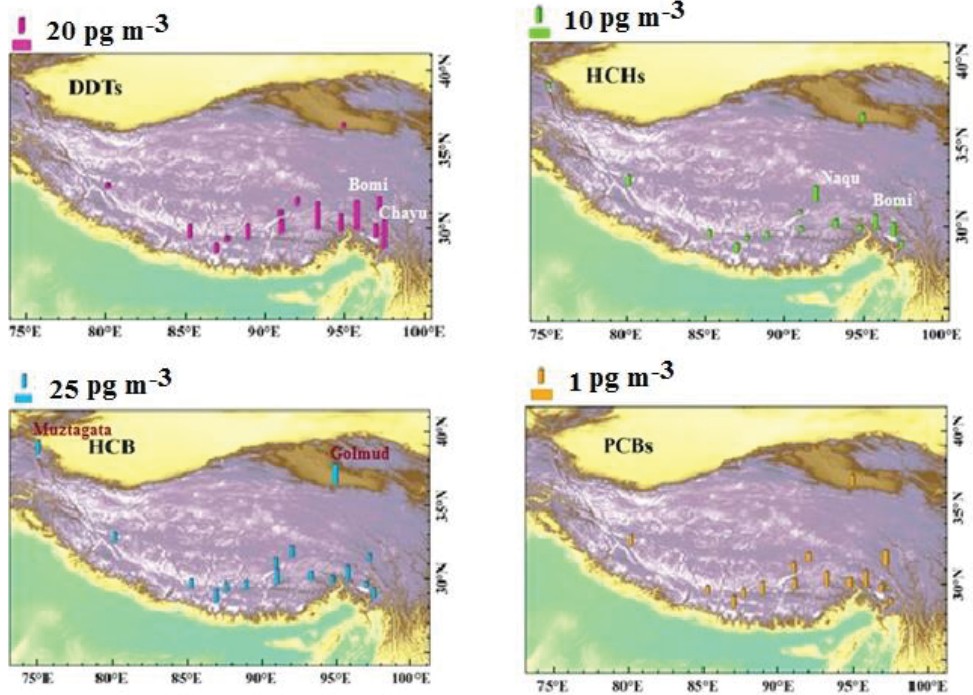

**Figure 1** Spatial distribution pattern of DDTs, HCHs, HCB and PCBs across the Tibetan Plateau



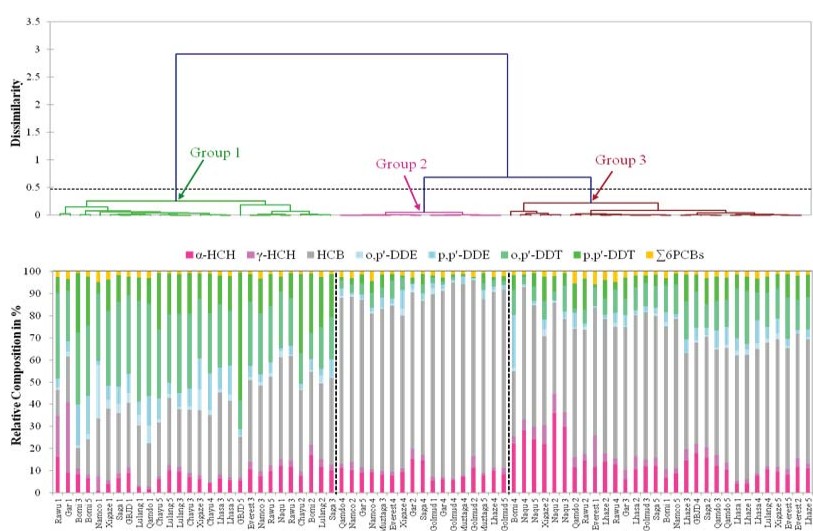

**Figure 2.** Dendrograph obtained by the cluster analysis (above) and the corresponding relative composition plot (bottom). The dendrogram includes a horizontal dashed line indicating where the number of clusters is stable, because moving up and down of this line along the similarity axis, the number of groups did not change. Samples in the composition plot were named by the name of sampling site and the Arabic number 1-5





which represent the sampling year from 2007 to 2012. For example, the 1$^{st}$ sampling year (from 2007 to 2008) was numbered as 1, and the 2$^{nd}$ year (from 2008 to 2009) was numbered as 2, and so on.





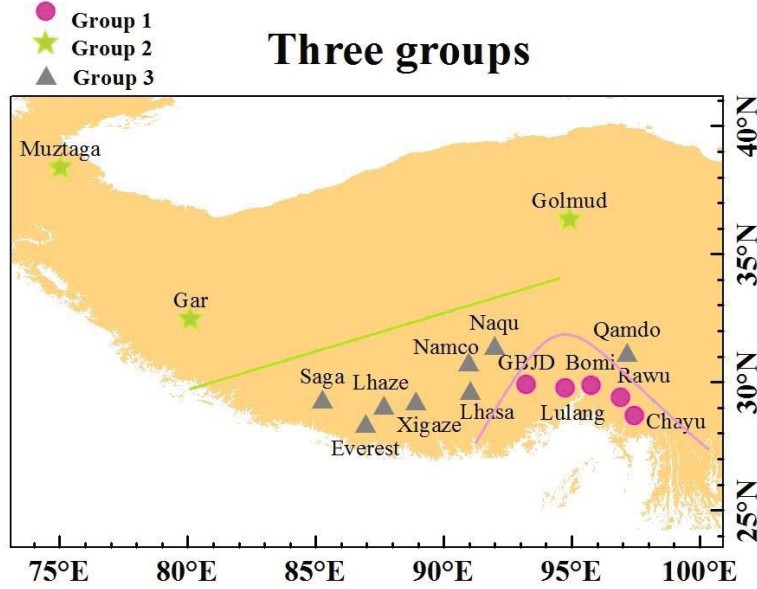

**Figure 3.** The geographic groups based on the cluster analysis (group 1: monsoon region,

group 2: westerly region, group 3: transition region)





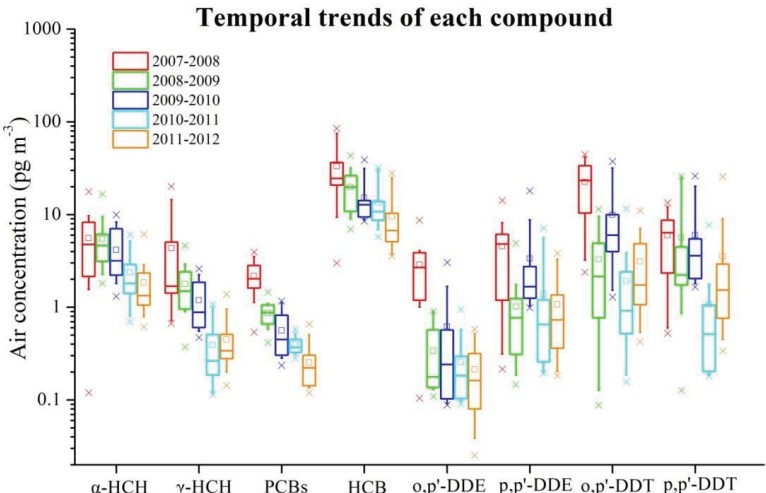

**Figure 4.**   The temporal trends of each compound over the five sampling years.