# Peer review of "Spatial Distribution of the Persistent Organic Pollutants across the Tibetan Plateau and Its Linkage with the Climate Systems: Five Year Air Monitoring Study"

_Atmospheric Chemistry and Physics, 2016_

## Referee Comment (RC1) · Anonymous Referee #1 · 1 Apr 2016

Review of "Spatial distribution of the persistent organic pollutants across the Tibetan plateau ant its linkage with the Climate systems: five year air monitoring study"

This is a well written and interesting paper reporting a relevant sampling effort of the POP concentrations in the Tibetan Plateau. The data set is novel and merits to be published. The limitations of the manuscript come from the fact of using passive samplers integrating the concentrations over one year. The authors make a poor job quantifying the uncertainty that this sampling approach introduce in the results. This is my only major comment to this manuscript, and it needs to be solved before publication. The

other minors issues can be found below.

- Page 2, Line 6. "climate interactions" is mentioned when the abstract does not deal with these!

- Page 2, Line 15. As the text says "are still ubiquitous", I suggest to update these references that are 10-17 years old!

- Introduction. Climate or weather fluctuations? Or both?

- Figure 1. There is no contrast of the bars over the map. I recommend a re-design of the map and figure in general.

- The passive samplers were deployed for 1 year! The authors argue that the obtained concentrations are the annual average. Is this true? Atmospheric concentrations of POPs do show a high variability depending on wind directions, season, etc. . ... If we focus on the influence of temperature, it is well known that gas phase concentrations show an inverse correlation with temperature. The gas-XAD partitioning coefficient is also temperature dependent. The influence of temperature is not linear (it is logarithmic), and therefore a passive sampling cannot get the average concentration over a long period of time with important temperature fluctuations. The influence of other variables can also be non-linear. The authors need to address these important issues, make the appropriate corrections, introduce an assessment of error propagation in their concentration estimates, and then discuss accordingly to the new results.

- Page 5, line 19. Storing XAD at -20C is a bad practice. Frezeeing will allow the humidity in the pores of XAD to break the XAD and introducing a blank problem.

- Re-assess the uncertainties, as 35% is too low for the sampling + estimation of sampled volumes + chemical analysis.

- Page 7, line 21. What is good agreement? Quantify this.

- Pages 7-8. To compare among sites and different studies in ng/sample does not make

any sense, unless the authors first demonstrate that the exposure time, the temperature and wind speeds are similar among the different samplers. I suggest to remove all this comparison, as it is useless. The only comparison possible is qualitative, and maybe on patterns.

- Page 9. The statements on the comparison of patterns should be supported by the appropriate statistical tests.

- Page 10. Which is the uncertainty on the fugacities in soil?

- Page 11. If air and soils are close to equilibrium, then it means that they are coupled in terms of concentrations. This means that the atmosphere supports and controls the soil concentrations of POPs, but that the soil support and control the atmospheric concentrations of POPs. This is obviously a secondary source. Equilibrium does not mean a zero flux, it just means that the volatilization flux is similar in magnitude to the depositional flux. A close air-soil coupling has also been observed using methods with lower uncertainty (Cabrerizo et al. EST 2013, Degrendele et al. EST 2016).

- Page 11. The lack of correlations between soil and air fugacity do not show anything, as the air concentrations are pseudo annual averages and not measured simultaneously with the soil samples. I would remove this paragraph.

- Obviously, the authors can not identify the seasonal variation, but as it may occur, does it affects the discussion?

- Page 14, line 21. I disagree that PAS are cheap and simple. It is simple to obtain a value for ng/sample, but very difficult to obtain a value for ng/m3 with an acceptable uncertainty that allows you to discuss certain patterns and processes. Therefore, both the "results"/cost and "new knowledge"/cost ratios may not be favorable neither.

- The paper relies too much on tables and Figures in the supplementary material. Sometimes is difficult to follow.

---

## Referee Comment (RC2) · Anonymous Referee #2 · 5 Apr 2016

General comments

Tibetan Plateau (TP) has been an idea platform for studying environmental fate of persistent organic pollutants (POPs). As it is lack from local contamination source for POPs in TP, the levels of different POPs can be significantly influenced by both Indian Monsoon air masses and European air masses. Thus, several Chinese national or international research groups have been engaging on TP for POPs study since a decade. In this work, a few classic OCPs and selected PCB congeners have been monitored across the TP from 2007 to 2012 based on an atmospheric POPs monitoring

program. The temporal trends of atmospheric POPs concentrations during this 5-year period have been discussed, and the spatial POPs patterns have been elucidated with the role of various climate systems as well. Overall, this manuscript is well written and the methods used sounds good. The findings are helpful to explore the environmental fate of POPs in TP. I recommend its acceptance after some revision.

Specific comments

One of the most significant developments in air sampling technology in recent years is the evolution of passive samplers. So far, there are several type of PASs have been developed for POPs study, such as PUF disk, PUF/XAD-4 combination, and XAD-2 cartridge. Have the authors already done any comparison, for example XAD-2 vs. PUF disk, before you chose XAD-2 for a five years program?

XAD-2 based PAS has been deployed at 16 background sites across TP for 5 years or even long. Although PAS has several advantages for spatial and long-term trends monitoring, the data quality have been always a concern for long-term deployment. The authors should be able to control or compare the PAS data with active sampler at one of these 16 stations. If there any data from an high-volume air sampler available, you may show them in the manuscript in order to prove the data quality generated from XAD-2 based PAS.

"The following compounds were measured and quantified: PCB 28, 52, 101, 138, 153 and 180; HCB; o,p'-DDE, p,p'-DDE, o,p'-DDT and p,p'-DDT; $\alpha$-HCH, $\beta$-HCH, $\gamma$-HCH and $\delta$-HCH" in this work, I am wondering how you selected these OCP species and PCB congeners. Are these compounds mostly partitioning in vapor phase or particle phase?

In the section of QA/QC, you reported that "Duplicate PAS were deployed to check the repeatability and the results showed the average relative deviation of concentrations between duplicates is generally low, which ranged from 9% to 18% for different compounds (Table S4). Uncertainties for chemical analysis were in the range of 25–35%

and reasons for the uncertainties were discussed in Text S3." I am wondering how you can achieve a 9% to 18% relative deviation for duplicate PAS when the uncertainties of analytical method ranging from 25-35%. Frankly, I may trust the data when you report relative deviation even higher than 50%, but it is a surprise that you achieved such good results for duplicate PAS.

In the section "3.1 Concentrations of POPs in the Atmosphere of TP" the authors used ng/sample for POPs concentration, while the unit was change to pg/m3 in the section "3.2 Spatial Distribution of POPs across the TP". I will suggest using the same unit to avoid any confusing understanding.

In the section, "3.3 Dose the soil-air exchange (secondary source) affect the spatial pattern?" the authors calculated air-soil fugacity for selected substance to evaluate that regional re-evaporation from soil can be an important vector for atmospheric POPs. Generally, atmospheric POPs data generated from PAS might be with very high uncertainties more than what we can expect. Fugacity calculated based on such data set is very hard to guaranty the model results. Moreover, as it is a five-year program, it will be very helpful if the authors have collected the soil samples from 16 monitoring sites, and then they use the data from paired air/soil samples for fugacity calculation.

Both Figure 1 and 2 should be improved for publication.
* * *

---

## Author Comment (AC1) · 16 May 2016

We are grateful to the reviewers' thoughtful comments and have now amended the manuscript according to their points. Some of these comments have been extremely helpful. We have acknowledged the valuable contribution made by the reviewers in this manuscript. A detailed response to each of the reviewers' points is provided below and we have carefully revised the manuscript as a result (all revisions are highlighted in the text). We hope the manuscript is now acceptable for publication in ACP.

1. **The limitations of the manuscript come from the fact of using passive samplers integrating the concentrations over one year. The authors make a poor job quantifying the uncertainty that this sampling approach introduce in the results. This is my only major comment to this manuscript, and it needs to be solved before publication.**
   **Response: In the revision, we discussed the uncertainties of chemical analysis and uncertainties of sampling rate. Please see page 6, line 12-20; page 7, line 1-18 .**
   **There are several factors involved here – how accurate are passives, and how reproducible are their results. A comparison of active and passive sampling as part of this manuscript implies that results agree within a factor of 2-3 between both sampling approaches. The reproducibility of numerous passives samplers co-deployed at one location is much better than that, typically less than 25 % (Table S3). This implies that passive air samplers can indeed be used to derive spatial and temporal trends.**

2. **Page 2, Line 6. "climate interactions" is mentioned when the abstract does not deal with these.**
   **Response:In page 1 line 27, we mentioned the "The interactions between the DDT polluted Indian monsoon air and the clean westerly winds formed a transition zone in central Tibet". Given the interaction between Indian monsoon and westerly winds were depicted using passive air samplers in our study, we infer that passive air samplers can indeed be used to investigate the interactions between POPs and climate.**

3. **Page 2, Line 15. As the text says "are still ubiquitous", I suggest to update these references that are 10-17 years old.**
   **Response: We changed these references. Please see page 2 line15.**

4. **Introduction. Climate or weather fluctuations? Or both?**
   **Response: Weather is the day-to-day state of the atmosphere, while, climate describes the variation of weather for a given place for a long time interval (decades). Both climate and weather fluctuations will lead to the variation of temperature, wind flow and precipitation that may finally influence the**

spatial distribution of POPs. In the introduction, we were not concerned with the short-term weather fluctuations. Instead, we focus on broad climate patterns, such as the Arctic Oscillation (AO), the North Atlantic Oscillation (NAO) and the Pacific North American (PNA) pattern.

5. Figure 1. There is no contrast of the bars over the map. I recommend a re-design of the map and figure in general.
   Response: We re-designed this figure. Please see page 27.

6. The passive samplers were deployed for 1 year! The authors argue that the obtained concentrations are the annual average. Is this true? Atmospheric concentrations of POPs do show a high variability depending on wind directions, season, etc. If we focus on the influence of temperature, it is well known that gas phase concentrations show an inverse correlation with temperature. The gas-XAD partitioning coefficient is also temperature dependent. The influence of temperature is not linear (it is logarithmic), and therefore a passive sampling cannot get the average concentration over along period of time with important temperature fluctuations. The influence of other variables can also be non-linear. The authors need to address these important issues, make the appropriate corrections, introduce an assessment of error propagation in their concentration estimates, and then discuss accordingly to the new results.
   Response: Passive air samples that operate in the linear uptake rate (infinite sink), such as XAD-passive samplers can be deployed for long time and yet accurately record ambient concentrations, as was demonstrated previously in several studies already: (1) Wania et al., developed the XAD-PAS and calibrate the uptake of XAD-PAS by simultaneously applying an active air sampler (AAS, Environ. Sci. Technol. 2003, 37, 1352-1359). They found the amount of chemical collected by XAD-PAS increased steadily over a 1-yr sampling period (Please see the Figure 1 below). (2) Gouin et al., conducted the field testing of XAD-PAS for the currently used pesticides (CUPs) in tropical region of Costa Rica (Environ. Sci. Technol. 2008, 42, 6625–6630), the length of deployment varied from 4 months to a year. They also obtained the linear uptake curves for CUPs (Please see the Figure 2 below). (3) We also did the field calibration of XAD-PAS in a sampling site (Lulang) of the Tibetan plateau (data not published); we also found the continuously linear uptake of various POPs on XAD-PAS (Please see Figure 3 below).

[Figure]

**Figure 1 Increase of the concentrations of several POPs measured in passive air samplers deployed for up to 1 yr at the three monitoring sites: Alert**
**Environ. Sci. Technol. 2003, 37, 1352-1359**

[Figure]

**Figure 2 Uptake curves of several current-use pesticides in XAD-PAS deployed for periods of up to one year at the field site in Belen, Costa Rica.**
**Environ. Sci. Technol. 2008, 42, 6625–6630**

[Figure]

[Figure]

**Figure 3. Uptake curves of HCHs, HCB, and DDTs in XAD-PAS in Lulang (data not published)**

Since both the previous studies and our own experiences observed the linear uptake of XAD-PAS to various target compounds, we do believe that XAD-PAS used in the present study provided the integrated ambient concentrations over one-year. If the PAS was operated in the linear uptake phase, it allowed us to obtain the time-averaged air concentration ($C_{air}$) over the entire sampling deployment (t) by using the following equation $C_{air} = \frac{m_{PAS}}{R*t}$.

Where, R represents the sampling rate (R) of the PAS and $m_{PAS}$ is the amount of chemical sorbed by the PAS.

Hayward et al., carefully compared the data provided by the conventional AAS and XAD-PAS over a one-year deployment. They compared the annually averaged concentrations obtained by AAS and XAD-PAS, and they found, for all pesticides, the data are not statistically different. Hayward et al., actually recommended using XAD-PAS to get the annual average of POPs because only minimal efforts should be paid as compared with conventional AAS (AAS usually collect samples once every two weeks, or operated continuously for each two week period ). Please see Environ. Sci. Technol. 2010, 44, 3410–3416.

Take all these evidences together, we think that the concentrations obtained in our study are good representations of their annual average.

7. Page 5, line 19. Storing XAD at -20C is a bad practice. Frezeeing will allow

the humidity in the pores of XAD to break the XAD and introducing a blank problem.

Response: We did not store XAD in freezer for very long time. Generally, after harvesting and transporting, we started the extraction after less than 20 days in freezer. Similar procedure has been done by the previous study (Krogseth et al., *Environ. Sci. Technol.*, 2013, *47* (9), pp 4463–4470).

Moreover, Table S1 provided the blank values, and fortunately, we did not get any blank problem.

Reference: Krogseth et al., Calibration and Application of a Passive Air Sampler (XAD-PAS) for Volatile Methyl Siloxanes. Environ. Sci. Technol., 2013, 47 (9), pp 4463–4470.

8. **Re-assess the uncertainties, as 35% is too low for the sampling + estimation of sampled volumes + chemical analysis.**

Response: We discussed two kinds of uncertainties in this study. The one is the uncertainty for chemical analysis and it was around 35%. Another uncertainty is from the sampling rate of XAD-PAS. We listed the reasons that may cause the uncertainties in the original version of text (line 24, page 6). In reversion, we added a separately paragraph to assess the uncertainties for the total processes (sampling rate + chemical analysis). Please see line 12-20, page 6 and line 1-18, page 7.

9. **Page 6, line 21. What is good agreement? Quantify this.**

Response: The difference between AAS and PAS is within a factor of 2-3. We re-organized this paragraph and please see line 1-12, page 7.

10. **Pages 7-8. To compare among sites and different studies in ng/sample does not make any sense, unless the authors first demonstrate that the exposure time, the temperature and wind speeds are similar among the different samplers. I suggest to remove all this comparison, as it is useless. The only comparison possible is qualitative, and maybe on patterns.**

Response: Agreed. We have deleted this paragraph.

11. **Page 9. The statements on the comparison of patterns should be supported by the appropriate statistical tests.**

Response: A new table listed the result of randomized block ANOVA to test differences between different sampling sites. Please see table S7.

12. **Page 10. Which is the uncertainty on the fugacities in soil?**

Response: Soil fugacity can be calculated by the following equation $f_s = C_s \rho_s RT / 0.41 \varphi_{om} K_{oa}$. Assuming an error of approximately 20% in $K_{OA}$ and 35% in analysis, this will result in a propagated error of approximately 55% in the

**soil fugacity.**

.

13. **Page 11. If air and soils are close to equilibrium, then it means that they are coupled in terms of concentrations. This means that the atmosphere supports and controls the soil concentrations of POPs, but that the soil support and control the atmospheric concentrations of POPs. This is obviously a secondary source. Equilibrium does not mean a zero flux, it just means that the volatilization flux is similar in magnitude to the depositional flux. A close air-soil coupling has also been observed using methods with lower uncertainty (Cabrerizo et al. EST 2013, Degrendele et al. EST 2016).**

    Response: We agree with this viewpoint. The sampler used by Cabrerizo et al. (2013) and Degrendele et al. (2016) is a fugacity sampler which needs electricity to operate. Due to the very limited electricity supply in the Tibetan Plateau, using such kind of sampler is not feasible. Apparently, fugacity sampler can provide more accurate exchange fluxes than the method we used in this study. The method using XAD derived air concentration and soil collected in 2008 will also cause relatively large uncertainties; we will include this discussion in revision. Meanwhile, we will recommend using fugacity sampler in future study (please see line 17-20, page 16 ).

    Reference: Cabrerizo et al. Climatic and Biogeochemical Controls on the Remobilization and Reservoirs of Persistent Organic Pollutants in Antarctica. Environ. Sci. Technol., 2013, 47 (9), pp 4299–430
    Degrendele et al., Diurnal Variations of Air-Soil Exchange of Semivolatile Organic Compounds (PAHs, PCBs, OCPs, and PBDEs) in a Central European Receptor Area. Environ. Sci. Technol., 2016, 50 (8), pp 4278–4288

14. **Page 11. The lack of correlations between soil and air fugacity do not show anything, as the air concentrations are pseudo annual averages and not measured simultaneously with the soil samples. I would remove this paragraph.**

    Response: We agree with reviewer. In the revision, we deleted this paragraph.

15. **Obviously, the authors can not identify the seasonal variation, but as it may occur, does it affects the discussion?**

    Response: Indeed, seasonal variations of chemicals were not included in this study. This study focused on the spatial distribution of POPs rather than seasonal variations of POPs. Actually, based on the continuous air monitoring by AAS, we investigated the seasonal variation of POPs in Lulang, a site especially influenced by Indian Monsoon (Sheng et al., Environ. Sci. Technol. 2013, 47, 3199−3208). We do observed the clear seasonal trend which could be attributed to the seasonally varied Monsoon

system and we found the chemical fingerprints in Lulang were very similar to that reported by Indian studies. The Sheng et al. study confirms that POPs' fingerprinting can be used as chemical tracer to track the sources of air mass, supporting the results of this study. The focus of this manuscript is the comparison of annual average and spatial patterns of POPs; seasonal trends are beyond the scope of this work.

Reference: Sheng et al., Monsoon-Driven Transport of Organochlorine Pesticides and Polychlorinated Biphenyls to the Tibetan Plateau: Three Year Atmospheric Monitoring Study. Environ. Sci. Technol., 2013, *47* (7), pp 3199–3208

16. Page 14, line 21. I disagree that PAS are cheap and simple. It is simple to obtain a value for ng/sample, but very difficult to obtain a value for ng/m3 with an acceptable uncertainty that allows you to discuss certain patterns and processes. Therefore, both the "results"/cost and "new knowledge"/cost ratios may not be favorable neither.

Response: We disagree. If we say "PAS are cheap", we choose AAS as the reference. For example, the cost for a XAD-PAS (including the sampling chamber and steel-less XAD cartridge) is ~60$, while that for the AAS is 4000$. Moreover, PAS can be deployed in truly remote region where electricity is absent. Although the PAS can only provide semi-quantitative concentrations of POPs, these data are still valuable, especially for those remote regions that POPs levels have not been reported before. So far, numerous studies obtained atmospheric distribution patterns of POPs at different spatial scales, i.e. global, regional and continental scale. Global Atmospheric Passive Sampling (GAPS) study is already widely known. Passive air sampling is becoming an easy and routine way for capturing both spatial and temporal patterns of POPs.

Based on the advantage of PAS, we suggest that "a simple and cost-effective PAS can yield valuable data on the realistic synoptic atmospheric interactions"

17. The paper relies too much on tables and Figures in the supplementary material.

Response: We moved some figures from supplementary material to the main text. Please see Table 1,2 and Figure 1,2,3,4 in main text.

---

## Author Comment (AC2) · 16 May 2016

We are grateful to the reviewers' thoughtful comments and have now amended the manuscript according to their points. Some of these comments have been extremely helpful. We have acknowledged the valuable contribution made by the reviewers in this manuscript. A detailed response to each of the reviewers' points is provided below and we have carefully revised the manuscript as a result (all revisions are highlighted in the text). We hope the manuscript is now acceptable for publication in ACP.

1. **One of the most significant developments in air sampling technology in recent years is the evolution of passive samplers. So far, there are several type of PASs have been developed for POPs study, such as PUF disk, PUF/XAD-4 combination, and XAD-2 cartridge. Have the authors already done any comparison, for example XAD-2 vs. PUF disk, before you chose XAD-2 for a five years program?**

   **Response: So far, different passive air samplers (PAS) have been designed, which allowed samplers to integrate ambient concentrations over time scales as short as hours/days or as long as weeks/months/years. Based on the field calibration of polyurethane foam (PUF)-PAS and XAD-resin PAS, Gouin et al., (*Environ. Sci. Technol.*, 2008, 42 (17), 6625–6630) recommended that PUF-PAS and XAD-PAS are suitable for obtaining atmospheric concentrations of POPs on the time scale of seasons and years, respectively. By coating XAD-4 onto PUF disk, the sorbent impregnated PUF disk (SIP) PAS is a modified version of the PUF-PAS. Months and seasons (3 months) of deployment are usually deployed for SIP-PAS.**

   **Measurements in remote areas like the Tibetan Plateau are especially challenging due to the lack of electricity and high sampling costs. Due to these reasons, we finally chose XAD-PAS as the target sampler. If we use PUF/SIP-PAS, the advantage is we can get the seasonal variations of POPs, but the disadvantage is we need to travel across the Tibetan Plateau many times to collect samplers, which increased the sampling cost very much. Considering that we already know that XAD-PAS can give reliable result for year-round sampling, using XAD-PAS will provide benefits for both reliable POPs data and economic sampling cost. Before we conduct our field sampling we did not compare the above-mentioned 3 types of PAS, because we already know the proper deployment time for each of them.**

   **Reference:** Gouin et al., Field Testing Passive Air Samplers for Current Use Pesticides in a Tropical Environment. Environ. Sci. Technol., 2008, 42 (17), pp 6625–6630

2. **XAD-2 based PAS has been deployed at 16 background sites across TP for 5**

years or even long. Although PAS has several advantages for spatial and long-term trends monitoring, the data quality have been always a concern for long-term deployment. The authors should be able to control or compare the PAS data with active sampler at one of these 16 stations. If there any data from an high-volume air sampler available, you may show them in the manuscript in order to prove the data quality generated from XAD-2 based PAS.

Response: Actually, in this study, both XAD-PAS and high-volume air sampler (AAS) were deployed in Lulang and details about the data comparison were provided in Text S3.

[Figure]

The DDTs derived from PAS were not very different from the corresponding AAS concentrations; whereas the low molecular weight OCPs, such as α-HCH and γ-HCH, showed some larger discrepancy (Figure above). This may be caused by different air masses being sampled and different adsorption characteristics for two kinds of samplers. Under these restrictions, the concentration variability within a factor of 2 -3 is deemed to be acceptable (Gouin et al., 2005). Therefore, the differences in the present PAS/AAS comparison (Figure above) were acceptable, which demonstrate that good agreement was found between the derived PAS concentrations and AAS results.

3. "The following compounds were measured and quantified: PCB 28, 52, 101, 138, 153 and 180; HCB; o,p'-DDE, p,p'-DDE, o,p'-DDT and p,p'-DDT; _-HCH, _-HCH, -HCH and _-HCH" in this work, I am wondering how you selected these OCP species and PCB congeners. Are these compounds mostly partitioning in vapor phase or particle phase?

Response: HCH isomers and DDT class chemicals are the most predominant

POPs species in Asia environment. HCH and DDT had been extensively used in China and India. Many studies demonstrated that fresh use of these two class chemicals are still occurring in some south Asia counties, such as India, Pakistan, Nepal and Thailand etc. The Tibetan Plateau is surrounded by China, India, Nepal and Pakistan, thus, its environment is most likely influenced by HCHs and DDTs that emitted from these surrounding countries. This is the reason why we mainly concerned about these chemicals. Regarding PCBs, PCBs (CB28, 52, 101, 118, 153, 138, and 180) were recommended for monitoring by the European Union Community Bureau of Reference and are also six ICES (International Council for the Exploration of the Sea) PCBs. These PCB congeners were selected as indicators of wider PCB contamination due to their relatively high concentrations in technical mixtures and their wide chlorination range (3-7 chlorine atoms per molecule). As the legacy POPs, HCHs ($\alpha$-HCH, $\beta$-HCH, $\gamma$-HCH and $\delta$-HCH), DDTs (o,p'-DDE, p,p'-DDE, o,p'-DDT and p,p'-DDT) and PCBs(CB28, 52, 101, 118, 153, 138, and 180) had been widely measured in other studies, which provided the opportunities for data comparison. All these are reasons why we chose these chemicals as target compounds.

High chlorinated PCBs have less volatility and mainly associate with particle phase, while low chlorinated PCBs are relatively volatile and present in gas phase. XAD-PAS are designed for mainly collecting POPs that normally dominate in the atmospheric gas phase. This is why high chlorinated PCBs were less detected in our study (PCB 28 and PCB 52) were regularly detected in samples, while highly chlorinated PCBs such as PCB 180 were never detected in samples).

4.  In the section of QA/QC, you reported that "Duplicate PAS were deployed to check the repeatability and the results showed the average relative deviation of concentrations between duplicates is generally low, which ranged from 9% to 18% for different compounds (Table S4). Uncertainties for chemical analysis were in the range of 25–35% and reasons for the uncertainties were discussed in Text S3." I am wondering how you can achieve a 9% to 18% relative deviation for duplicate PAS when the uncertainties of analytical method ranging from 25-35%. Frankly, I may trust the data when you report relative deviation even higher than 50%, but it is a surprise that you achieved such good results for duplicate PAS.
    Response:In this study, duplicate samplers were deployed in 3~6 sites for consecutive 5-year monitoring. For every sampling year, the RSD of duplicates (sampler a and b) were calculated and then these RSDs were further averaged for the corresponding compounds. Finally, all RSDs for this 5-year monitoring study were average and RSDs between duplicates ranged from 17% to 24% for different compounds (Please see Table S3 in supporting information). First, our previous statement (RSD ranged from 9% to 18%) got wrong. We feel very sorry for this. Second, for some sampling

year, RSD of individual compounds can reach up to 48% for ∑PCBs and 30-40% for DDT class chemicals. However, when take all 5-year into account, the averaged RSDs (from 17% to 24%) are still low. Basically, the analytical uncertainty is the major driver of difference between co-deployed PASs, highlighting the robust nature and simplicity of PASs.

5. In the section "3.1 Concentrations of POPs in the Atmosphere of TP" the authors used ng/sample for POPs concentration, while the unit was change to pg/m3 in the section "3.2 Spatial Distribution of POPs across the TP". I will suggest using the same unit to avoid any confusing understanding.

Response: Similar comments were also raised by reviewer 1#. Here, we deleted the comparison with unit of ng/sampler.

6. In the section, "3.3 Dose the soil-air exchange (secondary source) affect the spatial pattern?" The authors calculated air-soil fugacity for selected substance to evaluate that regional re-evaporation from soil can be an important vector for atmospheric POPs. Generally, atmospheric POPs data generated from PAS might be with very high uncertainties more than what we can expect. Fugacity calculated based on such data set is very hard to guaranty the model results. Moreover, as it is a five-year program, it will be very helpful if the authors have collected the soil samples from 16 monitoring sites, and then they use the data from paired air/soil samples for fugacity calculation.

Response: We agree with the reviewer's concern. As we discussed in response to reviewer 1, using a direct soil fugacity would have been preferable. Yet soil concentrations are not expected to vary quickly over time, so the current approach remains a good first estimation of air-soil gradients.

7. Both Figure 1 and 2 should be improved for publication.

Response: We re-organized these figures, please see page 27 and 30 .